# Controllable Synthesis of 1, 3, 5-tris (1H-benzo[d]imidazole-2-yl) Benzene-Based MOFs

**Sanjit Gaikwad [1],[†], Ravi Kumar Cheedarala [2],[†], Ranjit Gaikwad [1], Soonho Kim [3],\* and Sangil Han [1],\***

[1]    Department of Chemical Engineering, Changwon National University, Changwon-Si 51140, Korea; sanjitgaikwad29@jbnu.ac.kr (S.G.); ranjitgaikwad1989@gmail.com (R.G.)

[2]    Department of Materials Science and Engineering, Ulsan National Institute of Science and Technology (UNIST), 50 UNIST-gil, Ulju-gun, Ulsan 689-798, Korea; ravi@changwon.ac.kr

[3]    Industry-University Cooperation Foundation, Changwon National University, Changwon-Si 51140, Korea

\*    Correspondence: shkims@changwon.ac.kr (S.K.); shan@changwon.ac.kr (S.H.)

†    These authors contributed equally to this research work.

**Abstract:** The growing interest in metal–organic frameworks (MOFs) in both industrial and scientific circles has increased in the last twenty years, owing to their crystallinity, structural versatility, and controlled porosity. In this study, we present three novel MOFs obtained from the 1, 3, 5-tris (1H-benzo[d]imidazole-2-yl) benzene (TIBM) organic linker. The formed TIBM crystal powders were characterized by scanning electron microscopy (SEM) to estimate the morphology of the particles, powder X-ray diffraction (XRD) to confirm the crystal structure, Brunauer–Emmett–Teller (BET) method for structural analysis, and thermogravimetric measurements to examine the thermal stability. The TIBM-Cu MOF showed excellent $CO_2$ (3.60 mmol/g) adsorption capacity at 1 bar and 298 K, because of the open Cu site, compared to TIBM-Cr (1.6 mmol/g) and TIBM-Al (2.1 mmol/g). Additionally, due to the high porosity (0.3–1.5 nm), TIBM-Cu MOF showed a considerable $CO_2/N_2$ selectivity (53) compared to TIBM-Al (35) and TIBM-Cr (10).

**Keywords:** MOF; adsorption; $CO_2$ capture; porous material; solvothermal synthesis

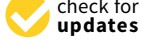



## 1. Introduction

The design and implementation of new porous materials for carbon dioxide ($CO_2$) separation by selective adsorption is a rapidly increasing research area, because of its importance in energy and environment-related applications [1–3]. The greenhouse effect responsible for global warming is mainly related to $CO_2$ emissions. Globally, $CO_2$ release is rapidly increasing because of the combustion of carbon-based fuels (oil, coal, and natural gas) and chemical reactions in petrochemical industries, steel, and cement [4]. Post-combustion $CO_2$ capture is considered the most effective technique for minimizing $CO_2$ emitted from industrial and energy-related sources. To this end, different technologies have been applied for $CO_2$ capture, such as membrane separation [5,6] liquid ammonia, and amine absorption [7,8] and adsorption [9,10]. Membrane separation is not suitable to large-scale applications because of its short lifetime, limited performance at low pressure, and poor stability in acid gas environments [11]. The absorption process has been extensively used in power plants, thanks to the high $CO_2$ selectivity promoted by amine functional groups; however, substantial downsides include equipment corrosion, high energy intake, and toxic ammonia loss [12].

Several materials have been investigated for their promising $CO_2$ capture and catalytic conversion properties, including supported silica [13,14], zeolites [15,16], metal oxides [17,18], bio-waste-derived carbons [19,20], metal–organic frameworks (MOFs) [21,22], porous carbons [23,24], and recently, a new class of porous materials, i.e., porous organic polymers (POPs). The last materials further encompass crystalline covalent organic frameworks (COFs) [25,26], triazine-based organic frameworks (CTFs) [27,28], conjugated microporous polymers (CMPs) [29,30], porous aromatic frameworks (PAFs) [31–33], and

hyper-cross-linked polymers (HCPs) [34,35]. Amongst the cited systems, MOFs have attracted significant attention, owing to their high surface area, pore volume, and surface functionality. Their unique crystal structure is composed of metal centers connected by organic linkers [36] that can be tuned to give rise to wide applications in gas storage [37–39], separation [39,40], drug delivery [41], and catalysis [42].

Generally, MOFs present high $CO_2$ adsorption capacities (2–5 mmol/g) at 298 K and high pressure, while the $CO_2$ capture capacity at below 1 bar is still challenging [42]. Therefore, many methods have been developed to obtain MOFs with improved adsorption capacity, such as metal doping in the metal–organic framework [43] and chemical functionalization with amine groups, such as polyethyleneimine (PEI) and tetraethylenepentamine (TEPA) [44–46]. Nevertheless, during the preparation, other aspects must be considered, such as recyclability, stability, and mild regeneration conditions. Researchers have highlighted alternative synthetic methods to modify MOFs stability, flexibility, and particle size [47,48]. The materials were prepared using a solvothermal method, which requires high-temperature treatment of metal salts and organic ligands in water or organic solvents [49]. Solvent properties, such as polarity and hydrophobic/hydrophilicity, affect the reaction mechanism. In particular, low-boiling-point solvents generate high vapor pressure under a reaction in a sealed reactor. The vapor pressure in a sealed reactor increases as the temperature increases, which causes the formation of a supercritical fluid, in which liquid and gas exist simultaneously. This supercritical-fluid phase improves the mixing of chemical reagents in the solvent, and causes product formation by crystallization, which does not normally occur. Hence, the use of low-volatility solvents increases the mobility of the dissolved ions and allows better mixing of the reagents [50–52]. The surface functionalities and structural topologies of MOFs are determined by various types of metal ions and the network-forming ligands. Concerning this, alteration in the functionality and structure of MOFs is achieved using different secondary building units (SBUs), specific functional groups, and multi-topic building blocks [53].

Several novel structures have been developed to improve the MOFs performance. Chen et al. proposed an NJU-Bai MOF with amide-functionalized carboxylate linkers and measured a significant improvement in $CO_2/N_2$ selectivity (166.7) at 298 K [54]. Wang et al. reported a covalent triazine framework (CTF) with N-heteroaromatic building blocks, which exhibited a $CO_2$ capacity of 3.48 mmol/g at 273 K and 1 bar [55]. Suresh et al. studied an amide-functionalized microporous organic polymer (Am-MOP) with a $CO_2$ capacity of 40 mL/g at 195 K and 1 bar. To improve $CO_2$ binding affinity, Zheng et al. introduced polar acylamide groups in the framework, and the obtained Porous Coordination Network (PCN) series MOF exhibited a $CO_2$ capacity of 20–24 mmol/g at 20 bar and 298 K [56]. Martin Schroder and coworkers prepared a series of isoreticular MFM MOFs functionalized with amide group, among which MFM-126 showed an excellent $CO_2$ adsorption capacity of 4.63 mmol/g at 298 K and 1 bar [56].

In this work, we report the preparation of novel MOFs consisting of 1, 3, 5-tris (1H-benzo[d]imidazole-2-yl) benzene (TIBM) as an organic linker and Al, Cr, and Cu as metal ions. The synthesis of the organic core integrated with imidazole rings, which possess secondary and tertiary amines, was crucial for the protonation of metal ions. In order to allow the correct metal incorporation and obtain high-quality materials, the imidazole rings of the ligand have to be equally distant and symmetrically anchored to the phenyl ring [57–62]. Additionally, the use of the TIBM linkers with different metal precursors affects the pore size and the $CO_2/N_2$ adsorption selectivity of the prepared system. In this regard, we synthesized three novel MOFs using different metal precursors (MOF-Al, MOF-Cr, and MOF-Cu) for $CO_2$ and $N_2$ capture. The obtained materials were examined using FT-IR, X-ray powder diffraction (XRPD), scanning electron microscopy (SEM), and thermogravimetric analysis (TGA). The volumetric method was used for $CO_2$ and $N_2$ adsorption capacity measurements.

## 2. Results

### 2.1. Characterization of the TIBM Based MOFs

#### 2.1.1. Morphological Properties

$N_2$ adsorption isotherms at 77 K (Figure 1a) were measured using a volumetric BET instrument (BELSORP-max, MicrotracBEL, Osaka, Japan) in order to characterize the structural properties of the MOFs. All the isotherms showed a typical Type I isotherm, corresponding to a microporous structure. The specific surface area and pore volume of the TIBM MOFs were determined by—-the Brunauer Emmett Teller (BET) method. TIBM-Cr presented the largest surface area and pore volume among the other MOFs (Table 1). The pore size distribution of the TIBM MOFs was determined by the non-local density functional theory (NLDFT) method. TIBM-Cr MOFs showed the largest pore size range (1.0–4.0 nm), compared to TIBM-Cu (0.3 to 1.5 nm) and TIBM-Al (1.0 to 3.0 nm) (Figure S1b in Supporting Information). The majority of pores appeared at 2 nm for TIBM-Al and TIBM-Cr, and at 1 nm for TIBM-Cu.

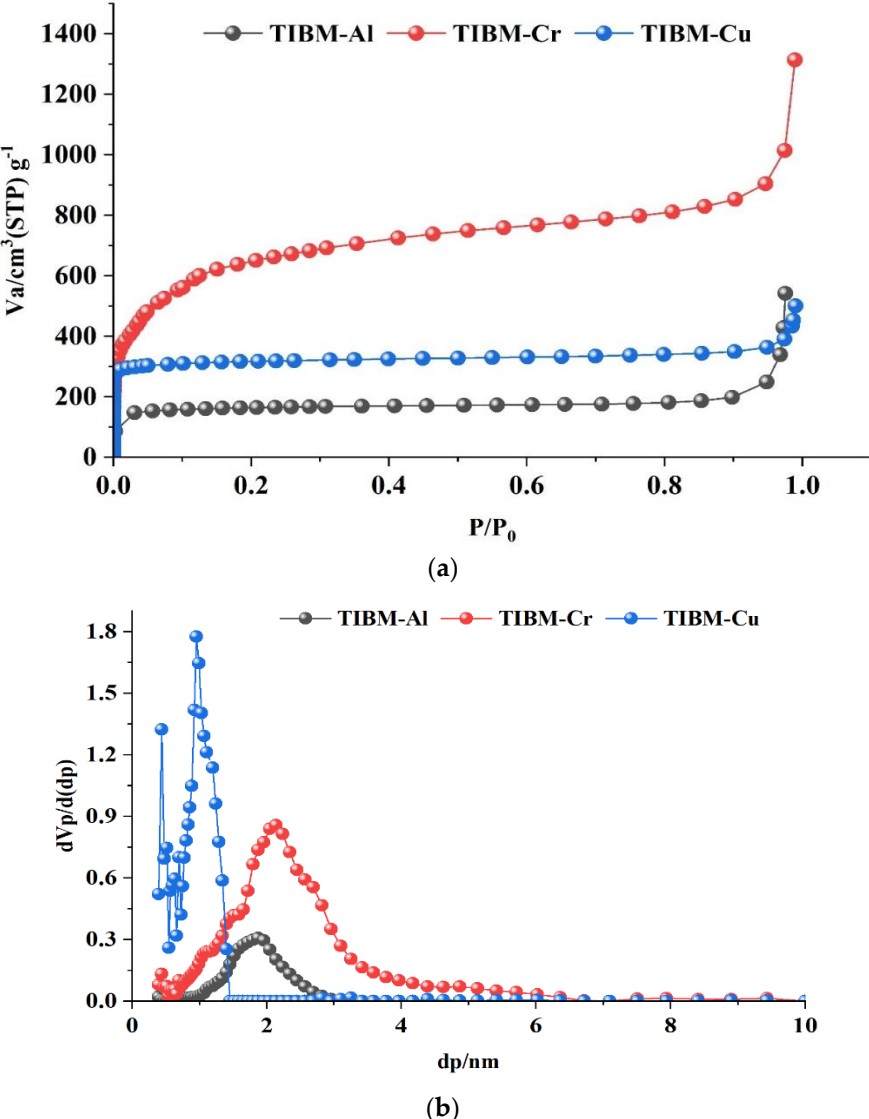

**Figure 1.** (**a**) $N_2$ adsorption isotherms and (**b**) pore size distribution of TIBM-Al, TIBM-Cr, and MOF-Cu at 77 K.

**Table 1.** Surface area and pore volume of TIBM-MOFs.

| Sample Name | BET Surface Area (m²/g) | Total Pore Volume (cm³/g) |
| --- | --- | --- |
| TIBM-Al | 505.19 | 0.681 |
| TIBM-Cr | 2141.1 | 2.116 |
| TIBM-Cu | 1073.5 | 0.778 |

Figure 2 reports the SEM images of the three TIBM MOF samples. The average particle diameters of the TIBM MOFs were determined using Image J software (NIH, Bethesda, MD, USA) based on the pixel distance of each image, which is correlated with the scale bar. TIBM-Cr exhibited a smaller particle size (0.25 μm) than TIBM-Cu (28.29 μm) and TIBM-Al (0.61 μm).

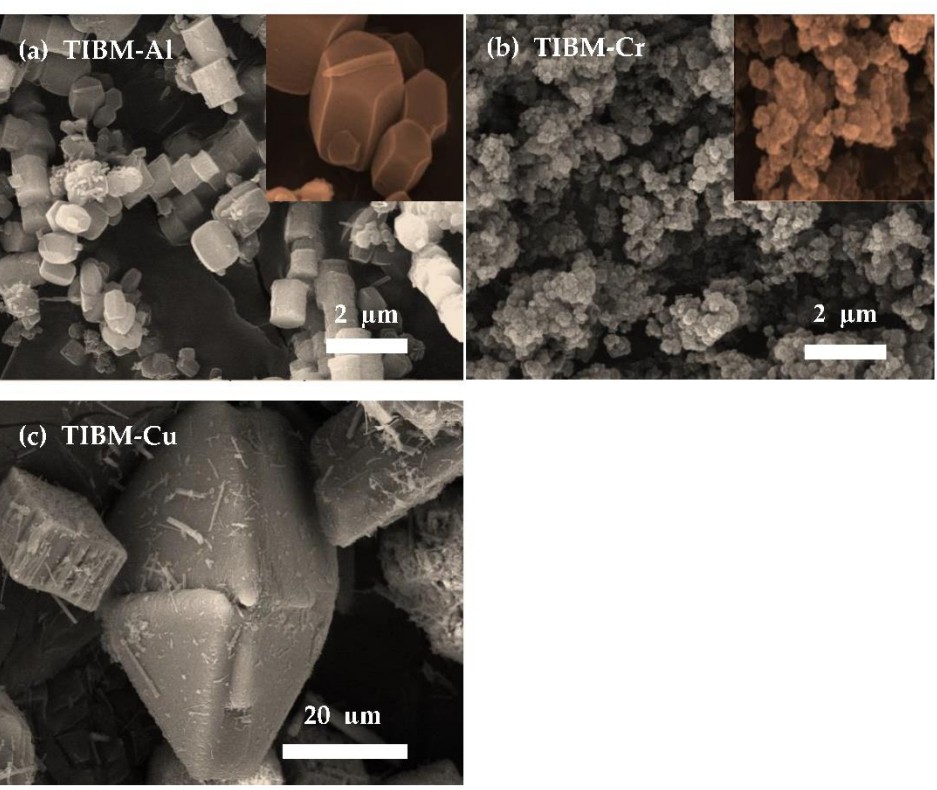

**Figure 2.** SEM images of (**a**) TIBM-Al, (**b**) TIBM-Cr, and (**c**) TIBM-Cu.

The MOF TIBM-Al showed a clear hexagonal crystal structure with sharp edges, spherical and highly porous nanoclusters were observed for TIBM-Cr, and a rhombic crystal structure was detected for TIBM-Cu. The TIBM-MOFs surfaces were found to be smooth, without evident cracks due to strong chelation between the central metal ion and TIBM amine linkers.

### 2.1.2. X-ray Diffraction (XRD) Analysis

The quality of TIBM-MOFs was optimized, and the samples were successfully doped with three metal ions by ionic-covalent bonds. The XRD pattern (Figure 3) of the TIBM-MOFs was compatible with the simulated pattern of UiO-66 from the Cambridge Structural Database (CCDC 837796) [63]. The patterns of TIBM-Cu and TIBM-Al are consistent with those of Cu-BTC and HKUST-1, respectively, while for TIBM-Cr, a resemblance to that of MIL-101 (Fe) and Fe-BTC was found [64–71], thus confirming the successful synthesis and integrity of the crystal structure after the coordination with TIBM (Table 2). In particular, the chelation was indicated by a broad peak and a sharp peak at 2θ values of 9° and 18°,

respectively, providing clear evidence of the crystalline nature of the TIBM-Al MOF. The Cr-TIBM MOF exhibited blunt peak intensities at 2θ values of 10° and 18°–20°, showing an increase in the crystalline degree of the MOFs [67,68]. On the other hand, the Cu-TIBM MOF showed sharp crystalline peaks, comparable to those of the Cu-based MOFs [64,65].

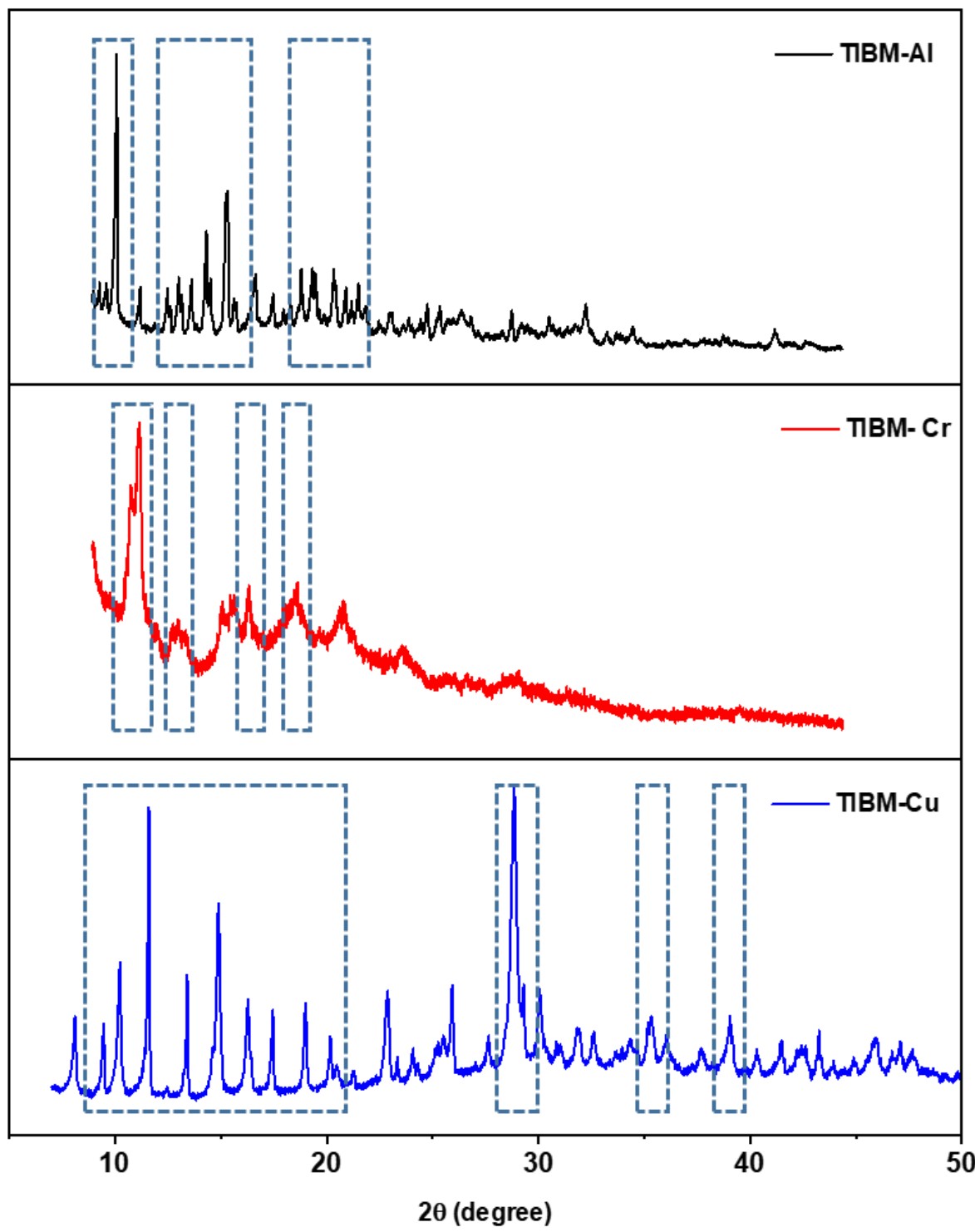

**Figure 3.** XRD patterns of MOF-Al, MOF-Cr, and MOF-Cu. (The dashed boxes represent the characteristic peaks that are matched with those of the reported MOFs).

**Table 2.** Matching the characteristics peaks of TIMB MOFs with reported MOFs.

| Synthesized MOFs | Matching Peak Angles (2θ) | Reported MOFs with Matching Peaks | Known Structure |
|---|---|---|---|
| TIBM-Al | 9.1, 12.5–20, 24–29 | Cu-BTC, Ni-BTC [67,68,72] | Hexagonal, cubic |
| TIBM-Cu | 9.4–20, 28.8, 35.3, 38.9 | Cu-BTC, HKUST-1 [64,65] | Cubic, pyramidal |
| TIBM-Cr | 10.9, 14, 20, 24 | Fe-BTC, MIL-100, MOF-235 [67–69,71,73] | Irregular spherical |

### 2.1.3. FTIR Analysis

The FTIR analysis allowed characterizing molecular interactions and bonding formation in the MOF frameworks (Figure 4). The strong stretching bands at 490–500 $cm^{-1}$ were attributed to metal-hydrogen bonds, particularly those of Cr and Cu metal ions. The bands at 720–724 $cm^{-1}$ and 750–754 $cm^{-1}$ correspond to =C-H bond modes in phenyl rings [74,75]. The characteristic MOFs bands, related to metal-ion-bound second and third amines (>NH-M-N and >N-M-N of MOF), appear in the range 1090–1100 $cm^{-1}$. Stretching bands due to C=C and C-H deformations of the phenyl rings were observed at 1399 $cm^{-1}$. The strong vibration mode at 1455 $cm^{-1}$ related to -NH and metal ions was attributed to the bidentate behavior of the N-M-N moiety. These characteristic peaks match well with the previously reported FTIR analysis of MOF-199 [28]. The strong resonance band exhibited by TIBM-Al was attributed to strong H-bonding of hydroxyl groups in the porous TIBM-Al material, as compared to TIBM-Cr and TIBM-Cu. This strong H-bonding occurs in Al-metal-based MOFs such as MIL-53(Al) and MIL-96 (Al), as compared to the Zn-based ZIF-8 and Zr-based UiO-66, and results in a strong resonance band [76–78]. The spectra for TIBM-Cr appear noticeably different after 3500 $cm^{-1}$, as compared to TIBM-Al and TIBM-Cu. The small peaks that were observed in the TIBM-Al and TIBM-Cu spectra after 3500 $cm^{-1}$ were attributed to the presence of crystalline water or the acidic -OH in carboxylic groups; these peaks do not appear in the case of TIBM-Cr [79,80]. The FTIR spectrum of the TIBM linker showed the typical peaks for NH wagging at 910 $cm^{-1}$, for C=C at 1420 $cm^{-1}$, for C=N at 1442 $cm^{-1}$, and 1610 $cm^{-1}$, for CO-NH at 1713 $cm^{-1}$, and for NH at 3448 $cm^{-1}$ (Figure 4b).

### 2.1.4. Thermal Analysis

During the TIBM MOFs thermogravimetric analysis (Figure 5), small drops in the range 50–150 °C were caused by dehydration. A further loss of 48% (300–500 °C) for TIBM-Cu, 28% (500–650 °C) for TIBM-Al, and 40% (400–600 °C) for TIBM-Cr, was ascribed to the decomposition of chelated imidazolium moieties [81,82]. According to the large drop temperatures, TIBM-Al presented better thermal stability than the other TIBM MOFs.

### 2.2. $CO_2$ and $N_2$ Adsorption Measurements

To examine the $CO_2$-capture performance of all the samples, $CO_2$ and $N_2$ adsorption properties were measured at 298 K and 0–1 bar. As reported in Figure 6, TIBM-Cu showed the highest $CO_2$ adsorption capacity (3.60 mmol/g) at 1 bar, compared to the TIBM-Al (2.04 mmol/g) and TIBM-Cr (1.67 mmol/g). This result is ascribable to both the metal site exposure and the metal oxide chelation of the TIBM host precursor for $Cu^{2+}$ ions [83]. The Cu-Cu magnetic interaction of TIBM-Cu is stronger than the Al-Al and Cr-Cr magnetic interactions of TIBM-Al and TIBM-Cr, respectively, and results in a strong interaction between $CO_2$ and the two available electrons of the $Cu^{2+}$ metal in the TIBM-Cu MOF [84,85]. Conversely, TIBM-Cr showed the highest surface area (2141.1 $m^2$/g) and total pore volume 2.116 $cm^3$/g, compared to the other examined MOFs (Table 1), while a reverse adsorption trend was found for $N_2$ adsorption for the TIBM MOFs. With regards to the $CO_2/N_2$ adsorption selectivity, which is defined as the ratio between $CO_2$ adsorption capacity at 0.15 bar and $N_2$ adsorption capacity at 0.85 bar, TIBM-Cu showed the highest value (53.69) compared to TIBM-Cr (11.16) and TIBM-Al (33.33), as reported in Figure 7, mainly in virtue of its high $CO_2$ adsorption capacity, and concurrent low $N_2$ adsorption capacity. In addition, $CO_2/N_2$ selectivity is different for each of the TIBM MOFs and depends on different factors,

such as MOF pore size and available open metal sites for $CO_2$. The pore sizes of TIBM-Cu (0.5 and 1 nm) are more suited to selective adsorption of $CO_2$ (0.33 nm) over $N_2$ (0.35 nm) than TIMB-Al (1.9 nm) and TIBM-Cr (2.2 nm), owing to the size-selective separation. The $CO_2/N_2$ adsorption selectivity of TIBM-Cu is promising, compared to other MOFs reported in the literature (e.g., 6 for CuDABCO [86], 8 for ZIF-8 [87], 12 for MIL-101 (Cr) [88], 18 for MOF-5 [89], 16.5 for MOF-177 [89]). However, the adsorption capacities of the TIBM MOFs are comparable to those of previously reported MOFs (Table 3). TIBM-Cu was selected for the performance evaluation, showing that the $CO_2$ adsorption capacity (Figure 8) gradually decreased with increasing temperature from 298 to 338 K.

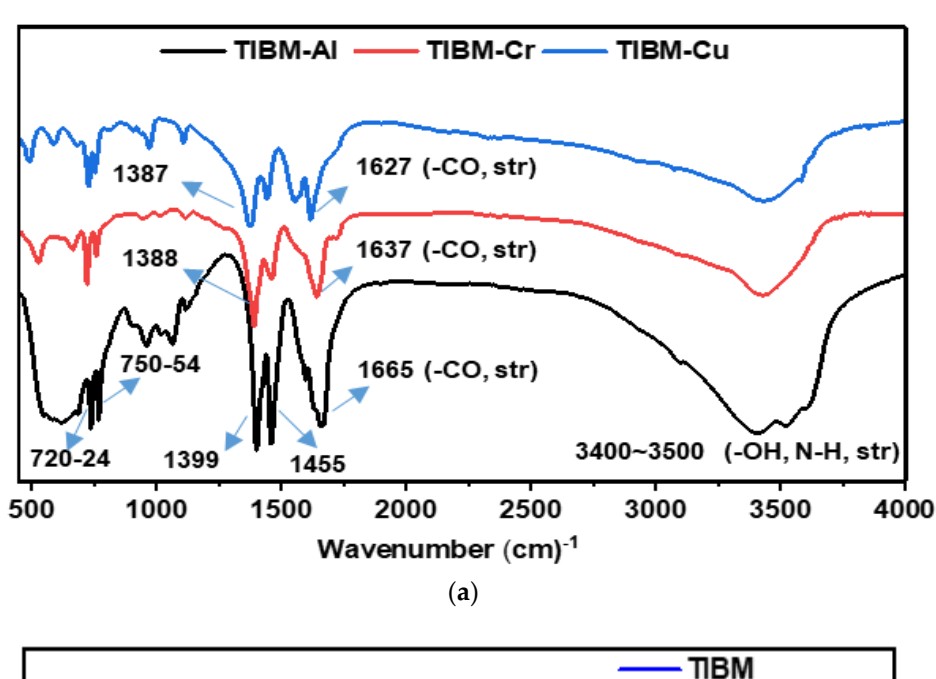

(**a**)

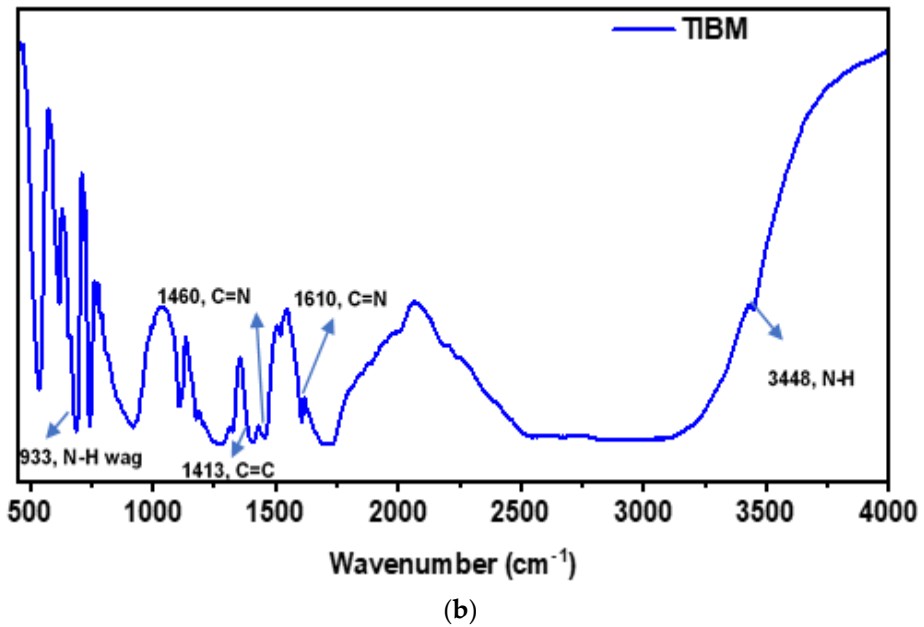

(**b**)

**Figure 4.** FTIR spectra of (**a**) TIBM−Al, TIBM−Cr, and TIBM−Cu, and (**b**) TIBM.

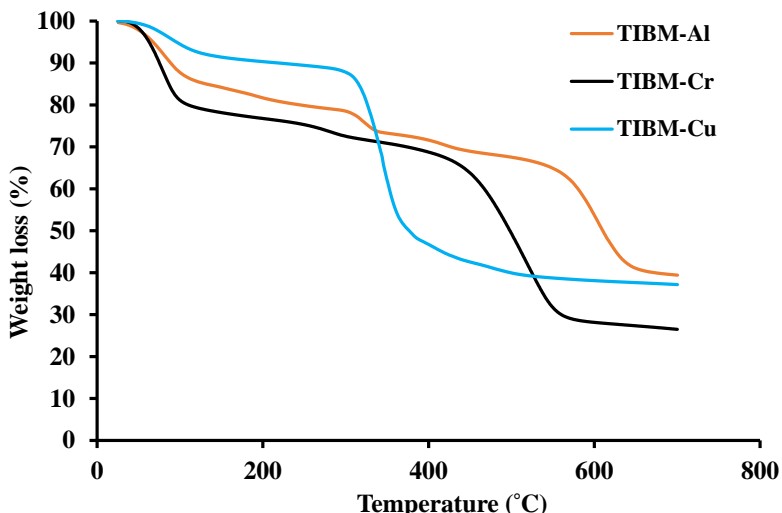

**Figure 5.** Thermal degradation graphs for TIBM-Al, TIBM-Cr, and TIBM-Cu.

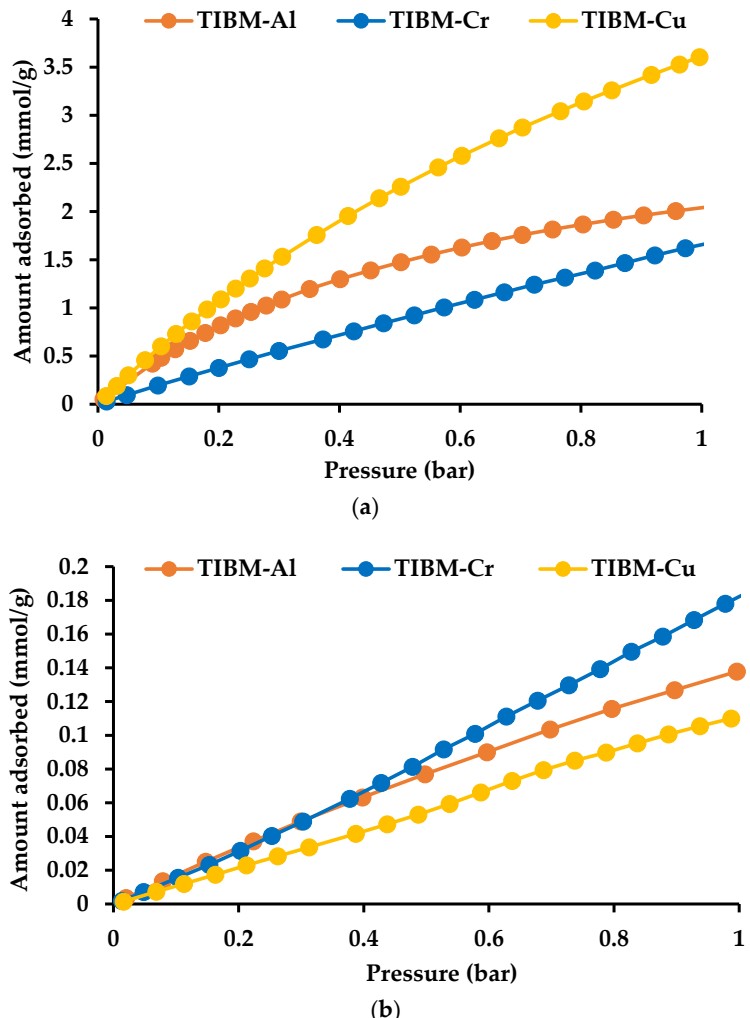

**Figure 6.** (**a**) $CO_2$ and (**b**) $N_2$ adsorption capacities at 298 K for TIBM-Al, TIBM-Cr, and TIBM-Cu MOFs.

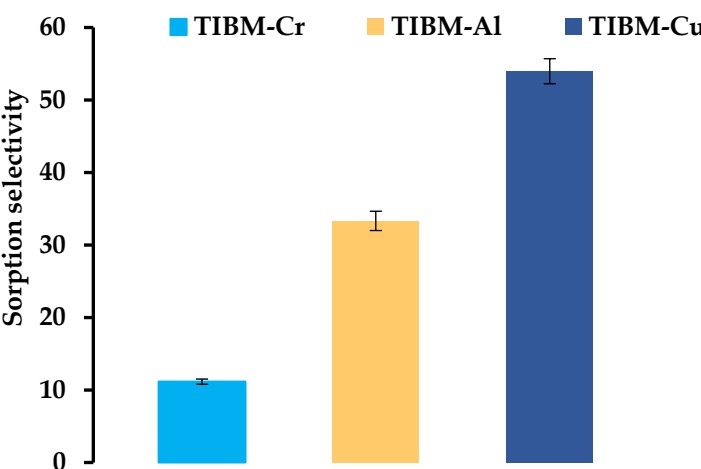

**Figure 7.** $CO_2/N_2$ selectivity for TIBM-Al, TIBM-Cr, and TIBM-Cu. Error bars represent the standard deviation (n = 3).

**Table 3.** $CO_2$ adsorption capacities of different MOFs.

| Adsorbent | Condition (Temperature/Pressure) | $CO_2$ Adsorption Capacity (mmol/g) | Reference |
|---|---|---|---|
| TIBM-Cu | 298 K/1 bar | 3.6 | This work |
| TIBM-Al | 298 K/1 bar | 2.0 | This work |
| TIBM-Cr | 298 K/1 bar | 1.6 | This work |
| IRMOF-74-III-$(CH_2NH_2)_2$ | 298 K/1 bar | 3 | [90] |
| SNU-5 | 195 K /1 bar | 2.6 | [91] |
| Fe-BTT | 298 K/1 bar | 3.1 | [92] |
| $[Cd_2L1(H_2O)]_2$ | 293 K /1 bar | 2.1 | [93] |
| $[Mg(3,5-PDC)(H_2O)]$ | 298 K/1 bar | 0.6 | [94] |
| $[Zn_4(OH)_2(1,2,4-BTC)_2(H_2O)_2]$ | 295 K/1 bar | 1.9 | [95] |
| $NH_2$-MIL-125 | 298 K/1 bar | 2.2 | [96] |
| IRMOF-74-III-$(CH_2NH_2)_2$ | 298 K/1 bar | 3 | [90] |
| IFMC-1 | 298 K/1 bar | 2.7 | [97] |
| TMOF-1 | 298 K/1 bar | 1.4 | [98] |
| MAF-23 | 298 K/1 bar | 2.5 | [99] |
| USTC-253 | 298 K/1 bar | 2.1 | [100] |
| $[Zn(L_2)]_n$ | 298 K/1 bar | 2.1 | [101] |
| USTC-253-TFA | 298 K/1 bar | 2.9 | [100] |
| SNU-71 | 298 K/1 bar | 1 | [102] |
| CPM-5 | 299 K/1 bar | 2.4 | [103] |
| SNU-M10 | 298 K/1 bar | 2.1 | [104] |
| UiO-66-$SO_3$H-0.15 | 298 K/1 bar | 2.2 | [105] |
| SHF-61 | 295 K/1 bar | 1 | [106] |
| Cu-BTTri-ens | 298 K/1 bar | 1.3 | [107] |
| SNU-31 | 298 K/1 bar | 0.6 | [108] |
| UiO-66-AD4 | 298 K/1 bar | 1.9 | [109] |

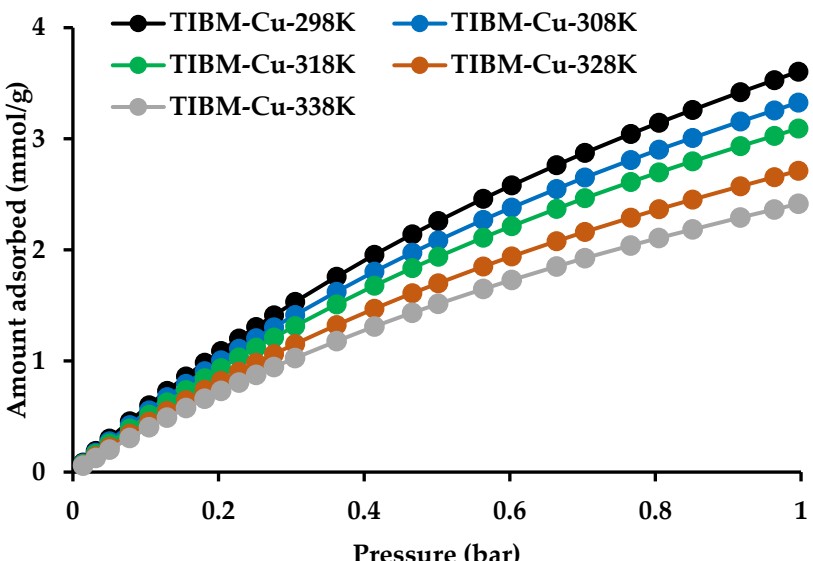

**Figure 8.** $CO_2$ adsorption capacity of TIBM-Cu MOF at increasing temperatures.

## 3. Materials and Methods

### 3.1. Chemicals

All chemicals were used as received, without further purification. Trimesic acid, chromium nitrate nonahydrate (99%), hydrofluoric acid (ACS reagent $\geq$ 48%), deionized water, copper acetate (trace metals basis $\geq$ 99.99%), ethanol (absolute alcohol $\geq$ 99.8%), aluminum chloride hexahydrate (reagent plus $\geq$ 99%), zinc acetate dihydrate (ACS reagent $\geq$ 98%), N,N-dimethylformamide (anhydrous, $\geq$ 99.8%), polyphosphoric acid (reagent grade), o-phenylenediamine ($\geq$ 98%), sodium bicarbonate solution (Bioreagent 7.5%), and methanol (ACS reagent) were purchased from Sigma Aldrich (St. Louis, MO, USA).

### 3.2. Preparation of 1, 3, 5-tris (1H-benzo[d]imidazole-2-yl) Benzene

O-phenylenediamine (7.2 g, 0.06 mol) was added to a solution of trimesic acid (4 g, 0.03 mmol) in polyphosphoric acid (PPA) (50 mL), and the reaction mixture was heated at 230 °C for 12 h (Figure 9). The resultant yellowish-colored reaction mixture was poured into ice-cold water (500 mL); upon stirring, the obtained brown precipitate was collected. The precipitate was neutralized by adding 20% sodium bicarbonate solution and filtered by centrifugation (4000 rpm). The brown solid converted into a white solid (82% yield) after recrystallization with methanol [110]; Mp. 280 °C. 1H NMR (400 MHz, DMSO-d6/TMS, ppm) δ 7.24–7.52 (m, 6H), δ 7.62–7.92 (m, 6H), 8.9 (s, 3H), 13.2 (s, 3H); $^{13}$C (100 MHz, DMSO-d$_6$/TMS, ppm) δ 115.2, 119.2, 122.2, 123.3, 125.5, 131.6, 135.8, 144.2, 159.1; ESI-MS: ([M + H]$^+$) m/z 427.18 (100%), found m/z 427.32 (100%), Chemical Formula: $C_{27}H_{19}N_6^+$, (Figure S1). The TIBM was synthesized as described in the literature [110]; the obtained pale yellow solid presented FT-IR and 1H-NMR spectra compatible with the ones previously reported [110,111].

### 3.3. Synthesis of TIBM-Cr

Chromium nitrate nonahydrate (0.6340 g) and TIBM (0.438 g) were dissolved in 20 mL deionized water by sonication for 10 min, and hydrofluoric acid (HF) (60 μL) was added to the mixture. The reaction mixture was transferred into a Teflon autoclave reactor sealed in a stainless steel vessel and maintained at 483 K for 48 h. The fine green-colored precipitate was washed three times in hot ethanol and five times in hot water. The final TBIM-Cr MOF was dried at 373 K and evacuated at 423 K for 12 h.

**Figure 9.** Synthesis of TIBM linker.

### 3.4. Synthesis of TIBM-Cu

Copper acetate (1.4 g) and TIBM (0.78 g) were dissolved in 30 mL water/ethanol (2:1) solution by sonication. HF (120 µL) was added to the mixture as a module. The reaction mixture was transferred into a Teflon autoclave reactor sealed in a stainless-steel vessel and kept at 423 K for 24 h. The blue-gray-colored precipitate was washed five times in ethanol and dried at 373 K, and then evacuated at 393 K for 12 h.

### 3.5. Synthesis of TIBM-Al

Aluminum chloride hexahydrate was dehydrated at 373 K for 10 h to remove water from the metal precursor. The dehydrated Al–metal precursor (1.2 g) and TIBM (0.626 g) were mixed in 30 mL water/ethanol (1:1) by sonication for 10 min. HF (60 µL) was added to the mixture as a module. The reaction mixture was transferred into a Teflon autoclave reactor sealed in a stainless-steel vessel, and maintained at 423 K for 48 h. The white-colored powder precipitate was washed five times in ethanol, subsequently dried at 373 K, and evacuated at 393 K for 12 h (Figure 10).

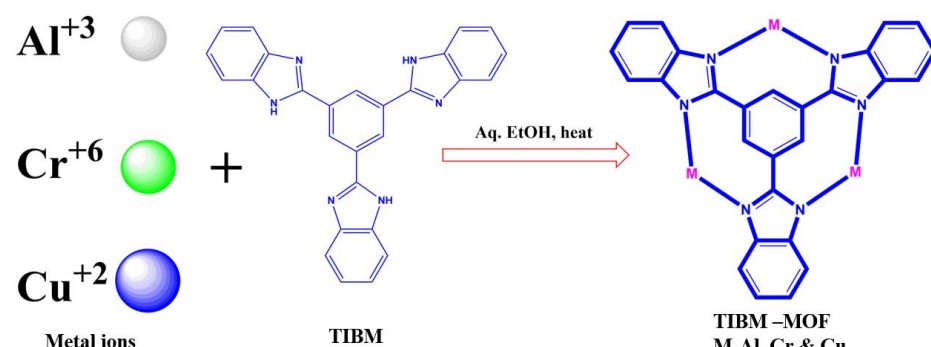

**Figure 10.** Synthesis of TIBM-MOFs (i.e., M= Al, Cr, and Cu).

### 3.6. Characterization

Fourier-transform infrared spectra (FTIR) in the range of 400–4000 cm$^{-1}$ were obtained with a Spectrum Two, PerkinElmer, UK spectrometer. A mixture of TIBM MOF power and KBr in the weight ratio of 1:99 was used to prepare the sample. The TIBM-MOFs particle morphology and crystal size were determined by means of SEM (Merlin compact, Carl Zeiss) at an accelerating voltage of 1 kV/10 kV, with a current of 10 µA. A volumetric method was used to analyze morphological properties, such as specific surface area, pore size distribution, and pore volume, using a BELSORP-mini (Microtrac BEL, Osaka, Japan) based on N$_2$ adsorption isotherm at 77 K. XRD analysis (X'Pert Pro-MPD, PANalytical, Almelo, The Netherlands) was performed to determine the TIBM-MOFs crystallinity.

For the thermal stability test, the samples were heated to 800 °C at a heating rate of 20 °C/min under $N_2$ atmosphere (50 mL/min), using a TGA instrument (Scinco TGA N1000, Twin Lakes, WI, USA). 1H-NMR spectra were obtained with a Bruker 400 MHz NMR spectrometer in $CDCl_3$, using tetramethylsilane (TMS) as an internal standard. Mass spectra were recorded on a Bruker Daltonik, Bremen, Germany, operated in linear mode with a pulsed nitrogen laser (337 nm, pulse frequency, 2 Hz).

*3.7. $CO_2$ and $N_2$ Adsorption Capacity Measurements*

$CO_2$ and $N_2$ adsorption on TIBM-MOFs were measured at 298–338 K at the pressure of 0–1 bar using a volumetric apparatus (BELSORP-mini, MicrotracBEL, Osaka, Japan). TIBM MOFs were first evacuated at 393 K (TIBM-Cu, TIBM-Al) and 423 K (TIBM-Cr) for 12 h, to remove impurities. A water circulating jacket connected to a thermostatic bath was used to control the measurement temperature with a precision of ±0.01 °C. A reference gas (helium) was used to determine the free space in the sample holder.

## 4. Conclusions

In summary, we have developed novel TIBM MOFs using a simple and inexpensive solvothermal process. The TIBM linker was prepared via condensation method, using trimesic acid and o-phenylenediamine. Interestingly, metal-modified TIBM MOFs (Cu-TIBM, Cr-TIBM, Al-TIBM) showed modifications in characteristic properties such as morphology, surface area, and pore size distribution. In particular, TIBM-Cu showed a remarkable $CO_2$ adsorption capacity (3.60 mmol/g) and selectivity (53), compared to TIBM-Cr and TIBM-Al. Due to the presence of open metal sites, N atoms of the imidazole functional groups, and an ideal pore size (0.3–1.5 nm) for selective $CO_2$ adsorption (the size of the pore aperture is similar to the size of $CO_2$), the TIBM-Cu $CO_2$ adsorption capacity is higher than that of previously reported MOFs, including MOF-5 (2.1 mmol/g), ZIF-8 (0.8 mmol/g), MIL-101 (Cr) (1.8 mmol/g) [112], MIL-101 (Cr, Mg) (2.0 mmol/g) [113], UiO-66 (2.5 mmol/g) [112], UiO-66- $NH_2$ (3 mmol/g) [114], and MOF-177 (0.8 mmol/g) [115,116]. Moreover, the TIBM-Cr MOF high surface area (2141 $m^2$/g) and pore volume (2.116 $cm^3$/g) points to the material potential for further applications.

**Supplementary Materials:** The following figures are available online at https://www.mdpi.com/article/10.3390/app11219856/s1, Figure S1: NMR spectra of TIBM linker.

**Author Contributions:** Writing—original draft preparation, conceptualization, and methodology, S.G. and R.K.C.; validation, S.G. and R.G.; supervision and writing—review and editing, S.H. and S.K.; investigation and resources, S.K. All authors have read and agreed to the published version of the manuscript.

**Funding:** This research was supported by the Changwon National University from 2021 to 2022.

**Data Availability Statement:** Not applicable.

**Acknowledgments:** This research was supported by the Changwon National University from 2021 to 2022.

**Conflicts of Interest:** The authors declare no conflict of interest.

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
