# Peer review of "Controllable Synthesis of 1, 3, 5-tris (1H-benzo[d]imidazole-2-yl) Benzene-Based MOFs"

_applsci, doi:10.3390/app11219856_

Round 1

Reviewer 1 Report

Title: Controllable synthesis of 1, 3, 5-tris (1H-benzo[d]imidazole-2-yl) benzene-based MOFs

The paper by Gaikwad et al. presents the synthesis and characterization of three new MOF systems obtained from the linker molecule 1,3,5-tris (1H-benzo[d]imidazole-2-yl) benzene (TIBM) using three different metal precursors. Aside from a thorough characterization, the authors investigated their application to CO2 and N2 adsorption and could demonstrate that, in particular, the Cu-based TIBM-MOF features an excellent CO2 adsorption capacity even at low pressure.

The paper is well structured, generally sound, and describes new CO2 adsorbing materials to the community. While the manuscript is well written, two sections need clarification (see below). I, therefore, recommend the submitted manuscript for publication in Applied Sciences after minor revision.

Minor remarks:

  1. Page 4, line 116: The authors report a hexagonal and rhombic crystal structure for the Al- and Cu-based MOF, but ‘porous nanoclusters’ in the case of Cr-based MOFs, which is very surprising. Seeing the XRD spectra for the three compounds, it seems that the Cr-MOF is polycrystalline or even amorphous. Do the authors reproductively find this behavior during synthesis?
  2. Page 4, line 120: ‘The average particle diameters … were determined using ImageJ…’ Please specify how you determined it (not only the software). Do you measure pixel distances and convert those into scales depending on the scale bar? Please specify.
  3. Page 4 line 122: TIBM-Cr showed a smaller particle size (0.25 nm) than TIBM-Cu (28.29 nm) and TIBM-Al (0.61 nm). Do you mean micrometer? How did you derive these values?
  4. Page 4 – Figure 2: Please remove the footer of each SEM image (which specifies the brand name of the SEM microscope) and label the panels with TIBM-Al/-Cr/-Cu in addition.
  5. Page 5 – Figure 3: What is the meaning of the blue dashed boxes? Please specify.
  6. Page 5 line 166: ‘The strong stretching bands at 490-500 cm-1 are attributed to the metal-hydrogen bonds.’ Why is there a strong resonance in the case of TIBM-Al and not for Chromium or Copper?
  7. Page 5 line 173: ‘The peaks at 3400-3500 cm-1 correspond to OH and second and third amine-groups of TIBM-MOFs.’ Please correct and rephrase this sentence. The wide peak around 3200-3600 cm-1 is due to the OH-stretch vibration of water, which was already assigned/detected at 1617 cm-1. The N-H stretch vibration of primary and secondary amines are between 3200-3400 cm-1 with two (symmetric/asymmetric for primary amines) and one resonance in the case of secondary amines. Tertiary amines don’t show resonances in this region. Hence, these might lead to additional modulations on top of the water signature. Why are they differently visible in the case of TIBM-Cr? What are the structures above 3500 cm-1?
  8. Page 6 – Figure 4: The IR spectrum of TIBM is not properly background corrected, nor correctly assigned. There should not be any absorption between 1700-2700 cm-1. Also, the assignment of the NH vibration at 3100 cm-1 is not visible. Please re-measure and replace the spectrum.
  9. Page 7 line 199: ‘The result is ascribable to both the metal site exposure and the metal oxide chelation of TIBM host precursor for Cu2+ ions.’ How do the authors know? Is it common knowledge (please add a reference in this case), or did the authors carry out additional experiments to identify the binding sites?
  10. Page 10 line 248: ‘… as described in literature’. Please cite the original publications.

Author Response

Reviewer 1

The paper by Gaikwad et al. presents the synthesis and characterization of three new MOF systems obtained from the linker molecule 1,3,5-tris (1H-benzo[d]imidazole-2-yl) benzene (TIBM) using three different metal precursors. Aside from a thorough characterization, the authors investigated their application to CO2 and N2 adsorption and could demonstrate that, in particular, the Cu-based TIBM-MOF features an excellent CO2 adsorption capacity even at low pressure.

The paper is well structured, generally sound, and describes new CO2 adsorbing materials to the community. While the manuscript is well written, two sections need clarification (see below). I, therefore, recommend the submitted manuscript for publication in Applied Sciences after minor revision.

Minor remarks:

  1. Page 4, line 116:The authors report a hexagonal and rhombic crystal structure for the Al- and Cu-based MOF, but ‘porous nanoclusters’ in the case of Cr-based MOFs, which is very surprising. Seeing the XRD spectra for the three compounds, it seems that the Cr-MOF is polycrystalline or even amorphous. Do the authors reproductively find this behavior during synthesis?

àWe appreciate reviewer comment, during the synthesis of TIBM-Cr, the similar behavior has been observed. Moreover, similar XRD patterns of TIBM-Cr have been reported for the MOFs such as MOF-100 (Cr), MOF-100, MOF-235 (Fe) [68-71] showing the presence of crystallinity and porous structure.

68.Torres, N.; Galicia, J.; Plasencia, Y.; Cano, A.; Echevarría, F.; Desdin-Garcia, L.; Reguera, E.J.C.; Physicochemical, S.A.; Aspects, E. Implications of structural differences between Cu-BTC and Fe-BTC on their hydrogen storage capacity. 2018, 549, 138-146.

69.Hindocha, S.; Poulston, S.J.F.d. Study of the scale-up, formulation, ageing and ammonia adsorption capacity of MIL-100 (Fe), Cu-BTC and CPO-27 (Ni) for use in respiratory protection filters. 2017, 201, 113-125.

70.Chen, M.-L.; Zhou, S.-Y.; Xu, Z.; Ding, L.; Cheng, Y.-H.J.M. Metal-organic frameworks of MIL-100 (Fe, Cr) and MIL-101 (Cr) for aromatic amines adsorption from aqueous solutions. 2019, 24, 3718.

  1. Simonsson, I.; Gärdhagen, P.; Andrén, M.; Tam, P.L.; Abbas, Z.J.D.T. Experimental investigations into the irregular synthesis of iron (iii) terephthalate metal–organic frameworks MOF-235 and MIL-101. 2021, 50, 4976-4985.

  1. Page 4, line 120:‘The average particle diameters … were determined using ImageJ…’ Please specify how you determined it (not only the software). Do you measure pixel distances and convert those into scales depending on the scale bar? Please specify.

àWith the help of pixel distance of each image, the average particle diameter was measured using ImageJ software. According to the comment, the below statement was added in the manuscript.

“The average particle diameters of the TIBM MOFs were determined using Image J software (NIH, USA) based on the pixel distance of each image, which is correlated with the scale bar. TIBM-Cr exhibited a smaller particle size (0.25 µm) than TIBM-Cu (28.29 µm) and TIBM-Al (0.61 µm).”

  1. Page 4 line 122: TIBM-Cr showed a smaller particle size (0.25 nm) than TIBM-Cu (28.29 nm) and TIBM-Al (0.61 nm). Do you mean micrometer? How did you derive these values?

àWe apologize for the mistake. It shoud be in micrometer, All values corrected to µm. We have calculated these values by considering pixel of each image using imageJ software, following sentence updated in the manucript in the line.

“TIBM-Cr exhibited a smaller particle size (0.25 µm) than TIBM-Cu (28.29 µm) and TIBM-Al (0.61 µm).”

  1. Page 4 – Figure 2:Please remove the footer of each SEM image (which specifies the brand name of the SEM microscope) and label the panels with TIBM-Al/-Cr/-Cu in addition.

à Thank you for the suggestions, the changes have been made in the manuscript.

  1. Page 5 – Figure 3:What is the meaning of the blue dashed boxes? Please specify.

àAccording to the suggestion, the meaning of the dashed boxes have been mentioned in the title of Fig. 3.

 “The dashed boxes represent the characteristic peaks that are matched with those of the reported MOFs”

  1. Page 5 line 166:‘The strong stretching bands at 490-500 cm-1 are attributed to the metal-hydrogen bonds.’ Why is there a strong resonance in the case of TIBM-Al and not for Chromium or Copper?

à “The strong resonance band exhibited by TIBM-Al was attributed to strong H-bonding of hydroxyl groups in the porous TIBM-Al material, as compared to TIBM-Cr and TIBM-Cu. This strong H-bonding occurs in Al-metal-based MOFs such as MIL-53(Al) and MIL-96 (Al), as compared to the Zn-based ZIF-8 and Zr-based UiO-66, and results in a strong resonance band [76-78].”

  1. Hadjiivanov, K.I.; Panayotov, D.A.; Mihaylov, M.Y.; Ivanova, E.Z.; Chakarova, K.K.; Andonova, S.M.; Drenchev, N.L. Power of infrared and raman spectroscopies to characterize metal-organic frameworks and investigate their interaction with guest molecules. Chemical Reviews 2020, 121, 1286-1424.
  2. Autie-Castro, G.; Autie, M.; Rodríguez-Castellón, E.; Aguirre, C.; Reguera, E. Cu-BTC and Fe-BTC metal-organic frameworks: Role of the materials structural features on their performance for volatile hydrocarbons separation. Colloids and Surfaces A: Physicochemical and Engineering Aspects 2015, 481, 351-357.
  3. Gopi, S.; Al-Mohaimeed, A.M.; Al-onazi, W.A.; Elshikh, M.S.; Yun, K. Metal organic framework-derived Ni-Cu bimetallic electrocatalyst for efficient oxygen evolution reaction. Journal of King Saud University-Science 2021, 33, 101379.
  4. Page 5 line 173: ‘The peaks at 3400-3500 cm-1correspond to OH and second and third amine-groups of TIBM-MOFs.’ Please correct and rephrase this sentence. The wide peak around 3200-3600 cm-1 is due to the OH-stretch vibration of water, which was already assigned/detected at 1617 cm-1. The N-H stretch vibration of primary and secondary amines are between 3200-3400 cm-1 with two (symmetric/asymmetric for primary amines) and one resonance in the case of secondary amines. Tertiary amines don’t show resonances in this region. Hence, these might lead to additional modulations on top of the water signature. Why are they differently visible in the case of TIBM-Cr? What are the structures above 3500 cm-1?

à  Following the comment, we removed the words of secondary and Tertiary amines. In general, the hydroxyl and amine groups will appear at the same region, the judgement is not accurate for those two functional groups, however, the -OH and NH groups assigned in the similar region with broad peak.

“The spectra for TIBM-Cr appears noticeably different after 3500 cm−1, as compared to TIBM-Al and TIBM-Cu. The small peaks that were observed in the TIBM-Al and TIBM-Cu spectra after 3500 cm−1 were attributed to the presence of crystalline water or the acidic -OH in carboxylic groups; these peaks do not appear in the case of TIBM-Cr [79,80].”

79 Salama, R.S.; El-Hakama, S.A.; Samraa, S.E.; El-Dafrawya, S.M.; Ahmeda, A.I. Adsorption, equilibrium and kinetic studies on the removal of methyl orange dye from aqueous solution by using of copper metal organic framework (Cu-BDC). Int. J. Modern Chem 2018, 10, 195-207.

80.Andonova, S.; Ivanova, E.; Yang, J.; Hadjiivanov, K. Adsorption Forms of CO2 on MIL-53 (Al) and MIL-53 (Al)–OH x As Revealed by FTIR Spectroscopy. The Journal of Physical Chemistry C 2017, 121, 18665-18673.

  1. Page 6 – Figure 4:The IR spectrum of TIBM is not properly background corrected, nor correctly assigned. There should not be any absorption between 1700-2700 cm-1. Also, the assignment of the NH vibration at 3100 cm-1 is not visible. Please re-measure and replace the spectrum.

àAccording to the reviewer suggestion, the spectra has been replaced with new spectra with correction of background. In addition, all peaks are correctly assigned and the N-H vibration is shown in 3448 cm-1.

  1. Page 7 line 199: ‘The result is ascribable to both the metal site exposure and the metal oxide chelation of TIBM host precursor for Cu2+ ions.’ How do the authors know? Is it common knowledge (please add a reference in this case), or did the authors carry out additional experiments to identify the binding sites?

àWe appreciate the reviewer comments, we didn’t perform additional experiments to identify the binding sites, as reviewer said it is common knowledge. The coordinated water molecule in the axial position of the paddlewheel Cu2+ centers of the structure can be removed by heating basically called as activation of MOFs. This activation creates potential active sites for gaseous adsorption.

According to the reviewer suggestions, following sentences were added in the manuscript with references.

“This result is ascribable to both the metal site exposure and the metal oxide chelation of the TIBM host precursor for Cu2+ ions [83]. The Cu-Cu magnetic interaction of TIBM-Cu is stronger than the Al-Al and Cr-Cr magnetic interactions of TIBM-Al and TIBM-Cr, respectively, and results in a strong interaction between CO2 and the two available electrons of the Cu+2 metal in the TIBM-Cu MOF [84,85].”

83.Peedikakkal, A.M.P.; Aljundi, I.H.J.A.o. Mixed-Metal Cu-BTC Metal–Organic Frameworks as a Strong Adsorbent for Molecular Hydrogen at Low Temperatures. 2020, 5, 28493-28499.

  1. Peng, Y.; Huang, H.; Zhang, Y.; Kang, C.; Chen, S.; Song, L.; Liu, D.; Zhong, C.J.N.c. A versatile MOF-based trap for heavy metal ion capture and dispersion. 2018, 9, 1-9.
  2. Ongari, D.; Tiana, D.; Stoneburner, S.J.; Gagliardi, L.; Smit, B.J.T.J.o.P.C.C. Origin of the Strong Interaction between Polar Molecules and Copper (II) Paddle-Wheels in Metal Organic Frameworks. 2017, 121, 15135-15144.

  1. Page 10 line 248: ‘… as described in literature’. Please cite the original publications.

à Following the comment, the statement has been cited by the original publication with reference.

Reviewer 2 Report

The researchers in this manuscript reported a methodology to realize controllable synthesis of three novel MOFs (Al, Cr and Cu) originated from TIBM organic linker and metal ions using solventhermal methodology. The researchers conducted a few routine characterizations on the MOF products, including FTIR, SEM, XRD, BET, TGA to obtain the micro and macro structural and thermal information for the new MOFs. Furthermore, they also compared the CO2/N2 adsorption selectivity of the three MOFs, and concluded TIBM-Cu has a better selectivity than the other two MOFs.

Overall, they described the synthesis paths, presented the characterization and adsorption test results, and addressed a considerable selective CO2 adsorption ability of the newly synthesized MOF at room temperature and atmospheric pressure which are very critical to evaluate MOF application in CO2 reduction.

The authors conducted a promising study, but lacked some in-depth critical discussions. Please see the reviews:

Majore reviews:

  1. In typical XRD analysis the major peaks and the represented crystal surfaces would be labelled to define the crystal structure of the materials. It would be great if the authors can do a more in-depth analysis or discussion on the crystal structures they indicated for the MOFs. Otherwise there would be little point of doing XRD characterization.
  2. In Figure 4(a), please re-plot the x-axis with a continuous and complete wavenumber range.
  3. In Figure 5, it would be great if the authors could explain why there is a big difference among the loss of imidazolium moieties. I understand this might not be critical to the overall scope of the manuscript, so I’ll leave the authors to decide if they want to add this discussion.
  4. When explaining the CO2 adsorption capacity advantage of TIBM-Cu, the authors explained “the result is ascribable to both the metal site exposure and the metal oxide chelation of TIBM host precursor for Cu2+ ions”. This is a vague explanation. What is this hypothesis or conclusion based on?
  5. It is good that the authors can successfully discover TIBM has promising CO2 adsorption capacity. However, overall the researchers do not show a convincing mechanism of the reason. Based on what they chose these three metal ions to compare (simply on that these are the most common ones?), and why they showed different adsorption selectivity? The manuscript ends up with an open question to this, which might be able to answer based on their characterizations. 
  6. Adding on to the above, material characterization purpose is not simply running through all the characteristics methods on the materials, but to extract critical information on the structural, and further functional properties. The authors gave moderately good results on the characterizations, but didn’t give enough discussions on the MOFs structures, and the adsorption mechanisms. I strongly recommend the authors could put more work on it. This will definitely elevate the whole research value of this study. 

Minor reviews:

  1. In the abstract (line 63 - 66), the author states: “Solvent properties, such as polarity and hydrophobic/hydrophilicity, affect the reaction mechanism. In particular, low-boiling-point solvents generate high vapor pressure under a reaction in a sealed reactor.”  Whereas the authors didn’t explain clearly the correlation between the vapor pressure vs the reaction mechanism. It would be more logic if the author added one or two sentences as a brief summary based on the previous studies.

Author Response

Please see the attached file for the reponse.

Round 2

Reviewer 2 Report

Thanks to authors' thorough response!